# Morphological Characteristics and Prevention of Tooth Enamel Demineralization during Orthodontic Treatment with Non-Removable Appliances

**DOI:** 10.3390/ijerph20010540

**Published:** 2022-12-29

**Authors:** Gagik Khachatryan, Marina Markaryan, Izabella Vardanyan, Mikayel Manrikyan, Gayane Manrikyan

**Affiliations:** 1Department of Dental and Pharmacological Professional Education, Yerevan State Medical University (YSMU), Koryun Str. 2, Yerevan 0002, Armenia; 2Department of Therapeutic Dentistry, Yerevan State Medical University (YSMU), Koryun Str. 2, Yerevan 0002, Armenia; 3Department of Pediatric Dentistry and Orthodontics, Yerevan State Medical University (YSMU), Koryun Str. 2, Yerevan 0002, Armenia

**Keywords:** enamel demineralization, non-removable appliance, prevention, white spot lesion

## Abstract

Despite the large number of studies on the effect of braces on teeth, there is no information on the dynamics of the state of the ultrastructure of the hard tissues of teeth during orthodontic treatment. The purpose of this study is to examine the state of the hard tissues of the teeth and carry out preventive measures to reduce the risk of complications in the process of orthodontic treatment using a non-removable device. Methods: For the in vitro study, 68 teeth were randomly divided into group A—no prophylactic treatment, and group B—treated with the fluorine varnish Tiefenfluorid. After 35 days, all the teeth were prepared for microscopic examination. The clinical study included 59 patients aged 12–17 years with orthodontic brackets. The split-mouth technique was used. The areas around the bracket of one-half of the oral cavity were treated with Tiefenfluorid every 6 months during the entire treatment period. The teeth of the second half of the oral cavity served as the control group. The data were processed in the SPSS19 package. Results: In vitro and clinical study results showed a statistically significant difference between the prophylactic and control groups of teeth in favor of the prophylactic group, where the statistical significance was *p* ≤ 0.01.

## 1. Introduction

With the enlargement in the number of dentoalveolar anomalies, there has been a corresponding increase in the need for orthodontic treatment by means of various types of non-removable devices. Its use is effective for the treatment of anomalies in the position of individual teeth and their groups, as well as occlusion anomalies. In the course of orthodontic treatment over a prolonged period, various non-removable devices are installed in the patient’s oral cavity, which are followed by a change in homeostasis that has a negative effect on the dentoalveolar system’s tissues and organs. The mastication function and the physical state of the dentoalveolar system change, and the quality of life of patients undergoing orthodontic treatment decreases [1,2]. The hygienic state of the oral cavity in the presence of a bracket system worsens due to various structural elements of the devices which serve as retention points for the accumulation of soft microbial biofilm, while the susceptibility of teeth to caries increases and inflammation develops in the periodontal tissues [3]. The impairment of the dynamic balance of metabolic processes, an increase in the number of pathogenic and opportunistic microorganisms that occur in the process of orthodontic treatment, and the development of serious complications against this background remain relevant. During orthodontic treatment, enamel demineralization is most often detected in the form of “white” spots but only at the stage of removing the orthodontic device [4]. A number of authors have studied the state of tooth enamel during orthodontic treatment on the bracket system. Notably, researchers have provided evidence that enamel damage after the removal of braces was quite significant [5,6]. Similar data were obtained by other authors [1,7].

The first signs of enamel demineralization appear 1 month after fixation of the orthodontic device and manifest as white spots that extend to a depth of 100 microns [8,9,10], which, in turn, can lead to further spread of the carious process if rational oral hygiene measures are not taken. White spot lesions (WSL) develop rapidly and often become an aesthetic problem for patients even years after the orthodontic appliances have been removed. However, their frequency has been reported to be widely variable, from 2% to 97%, in different epidemiological studies [2,9], which might be explained by the techniques used to detect and characterize them, including visual inspection and photographs [11]. In recent years, fluorescent diagnostics using DIAGNOdent devices (KaVo, Germany) for detecting initial caries has become more widespread [12,13].

When treated with non-removable devices, the tooth enamel is subjected to additional mechanical impact of orthodontic forces locally. Numerous methods are used for the prevention of caries using various organic and inorganic fluorine compounds: sodium fluoride and potassium; stannous fluoride; amino fluoride; monofluorophosphate; zirconium fluoride; and others [14,15,16,17,18]. One of these methods is deep fluorination, which is based on the use of new-generation fluorine-containing solutions that promote the formation of microcrystalline CaF_2_, which releases fluorine for a long time (more than 1 year) [19,20,21]. Effective prevention, diagnosis, and treatment of lesions will minimize caries formation and also provide an aesthetic smile. The most appropriate, cheapest, and easiest-to-apply treatment method should be preferred for the patient.

Despite the large number of studies devoted to examining the effect of non-removable orthodontic devices on changes in the tissues of the dentition, there is no information on the dynamics of the state of the ultrastructure of the hard tissues of the tooth at the stages of orthodontic treatment and related complications. Split-mouth randomized controlled trials (RCTs) are popular in oral health research. Split-mouth CTs have the advantage that most of the variability among patients is excluded from the assessment of the effect of the intervention to potentially increase statistical power, with each subject being its own control [22,23,24,25]. The purpose of this research is determined by the necessity to study the state of hard tissues of the teeth and carry out rational preventive measures to reduce the risk of complications in the process of orthodontic treatment using a non-removable device.

## 2. Materials and Methods

### 2.1. Study Design

The studies were carried out at the Dental Academy clinic and at the Department of Pediatric Dentistry and Orthodontics of YSMU. The scientific work consisted of clinical and laboratory studies. For the in vitro *study*, 68 teeth of different groups were used (mainly the first or second premolars of the upper or lower jaws, or the central and lateral incisors in the jaw, Figure 1). These teeth were removed according to orthodontic instructions. *The clinical study* was conducted with 59 orthodontic patients aged 12–17 who were treated with non-removable orthodontic appliances.

### 2.2. In Vitro Study

For the laboratory study, the extracted human teeth were used as a model for simulating ex vivo procedures. Human tissues are the first choice of material for evaluating the bond strength. To obtain comparable results, the samples were stored under certain uniform conditions. The antimicrobial properties of the preservation medium are rather critical as extracted teeth present a risk of cross-infection. The sterilization process should be neutral for the microstructure of enamel and dentine, as even a slight change can affect the adhesive bond [26,27]. The extracted teeth were directly stored in a 0.1% thymol solution and then each tooth was treated with 0.12% chlorhexidine for disinfection. Residual soft tissues were removed with a surgical lancet. The enamel surface of all the teeth was treated with a polishing abrasive (not containing fluorine) and distilled water, using circular brushes at a slow speed.

Next, the teeth were washed with distilled water and randomly divided into two equal groups (A and B). A small window was cut out of the adhesive tape according to the size of the bracket base. The tape was superimposed on the vestibular surface of the teeth and the window was located around the bracket. This was carried out to isolate tooth enamel from the acid treatment to avoid unintended demineralization. The part of the enamel that was within the adhesive tape was treated with 30% phosphoric acid for 30 s. After this process, the enamel was washed with distilled water for the same amount of time and thoroughly dried with a stream of air. After these procedures, metal braces manufactured by the Italian company Leone (Florence, Italy) were installed on all the teeth using special Brace Paste^®^ Adhesive (American Orthodontics, Sheboygan, WI, USA).

After the installation of braces, the teeth of the first group A were not subjected to prophylactic treatment. The teeth of the second group B were treated with an enamel-sealing liquid Tiefenfluorid (Humanchemie GmbH, Alfeld, Germany) [20,21,28], with 2 mm around the base of the bracket. Then, the teeth of each group were placed in separate containers that contained a solution of artificial saliva (20 µmol/L NaHCO_3_, 3 µmol/L NaH_2_PO_4_, and 1 µmol/L CaCl_2_ under neutral pH conditions).

### 2.3. Artificial Caries Induction 

After 12 h, the teeth were twice placed for 1 h into 200 mL of artificially demineralizing (caries-stimulating) solution (2.2 µmol/L Ca^2+^, 2.2 µmol/L PO_4_, 50 µmol/L acetic acid under conditions pH 4.4) with an interval of 6 h. Both solutions were kept in an incubator at a temperature of 37 °C corresponding to normal oral temperature and were renewed every 3 days during the 35 days of the study. All the teeth were removed from the solution, rinsed with deionized water for 5 s, and then brushed without the use of abrasives. Thus, the teeth of both groups were brushed twice a day. The teeth were transferred into two solutions (an artificial saliva solution and a caries-stimulating solution), adhering to the above rules for all 35 days. The teeth of the second group B on the 15th day were additionally treated with Tiefenfluorid. The transfer to solutions was carried out to obtain similar conditions in the clinical environment. After 35 days, all the teeth were removed from the solutions and dried; the brackets were removed, washed with distilled water, and then stored in boxes with distilled water.

### 2.4. Morphological Study

The preparation of slides for microscopic examination was carried out in the following order:Decalcification for 3 days in 5% nitric acid solution.Washing with running tap water during the day 5–6 times to remove the acid solution.Washed tissues were transferred to alcohol solutions of increasing concentrations for the dehydration procedure.After dehydration, a cleaning procedure was carried out by soaking in a xylol solution.The cleaned tissues were transferred into soft paraffin (melting point 46–48 °C).Then they were blocked by pouring into hard paraffin (melting point 56–58 °C).Preparation of 5 µm, 7 µm, and 10 µm thick sagittal sections taken from blocked tissues by means of a rotary microtome (Leitz, Wetzlar, Germany).Staining with Ehrlich’s hematoxylin for 5–10 min.After staining, the sections were dehydrated and covered with a coverslip [26].

The preparations were examined under the NIKON Eclipse Ci-L electron microscope (Nikon, Japan) at ×100 or ×200 magnification. In the part being examined in the area of the bracket, the depth and surface area of the demineralized enamel layer were measured. All examinations were performed using the same microscope to avoid possible differences between instruments. The average indicators for each group were calculated and a comparative analysis was carried out between the indicators of the groups.

### 2.5. Exclusion Criteria of Clinical Study

Exclusion criteria were used only in the clinical study, there were several:The presence of white spots on the teeth before fixing the braces.Refusal of the patient or parents to participate in the study.Ineligibility for the studied age groups.

### 2.6. In Vivo Study

Participation in this study was voluntary. All participants signed a consent form in accordance with the Declaration of Helsinki. The split-mouth technique [22,23] was used. Immediately after fixing the brackets, the areas around the base of the bracket on the teeth of one-half of the oral cavity were treated with an enamel-sealing liquid, Tiefenfluorid [21,29]. The teeth of the second half of the oral cavity served as a control group. The treatment was carried out every 6 months during the entire treatment period. The presence of a white spot on the surfaces of the teeth was determined. The demineralization and remineralization processes of WSL were studied using laser-fluorescence DIAGNOdent [12,13,30].

### 2.7. Statistical Analysis

Descriptive analyses (mean ± SD for continuous variables and frequencies/proportion for categorical variables) were computed for all variables of interest. Differences between the two groups were evaluated using the “T-Test” for categorical variables. *p*-value was considered significant at <0.05 and <0.001 for highly significant results. Analyses were conducted using the SPSS19 (SPSS Inc., Chicago, IL, USA) software.

## 3. Results

### 3.1. Results of Studies In Vitro

By the end of the study, tooth loss was observed in both study groups: two each in the A (control) and B (prophylactic) groups. The loss is explained by the damage they suffered during the manufacture of microscopic preparations. In each group, 32 teeth remained. In the control group, the proportion of premolars was 81.25%, and for incisors, it was 18.75%. In the prophylactic group, the proportion of premolars was 75%, and for incisors, 25%.

The average depths of decalcification in the control and prophylactic groups are shown in Figure 2. The results obtained are statistically significant: t = 11.7, *p* < 0.0001 in the control group and t = 10.02, *p* < 0.0001 in the prophylactic group.

The area of the lesion in the groups was also different (Figure 3) and statistically significant: t = 17.2, *p* < 0.0001 in the control group and t = 11.4, *p* < 0.0001 in the prophylactic group.

### 3.2. Results of Morphological Study

When discussing the study results, the obtained sections showed a statistically significant difference between the prophylactic and control groups of teeth in terms of the depth of the lesion and the surface area (t = 5.3, *p* < 0.0001 and t = 2.4, *p* < 0.02, respectively). Thus, decalcification is more pronounced in the first (control group), where the average thickness of the affected enamel layer is 1.83 times greater than in the first one (Figure 4 and Figure 5).

### 3.3. Clinical Results

The study involved patients aged 12 to 17 years who were divided into two groups: 12–14 years (32 patients, group C—control, and group D—preventive), and 15–17 years (27 patients, group E—control, and group F—prophylactic).

According to the preliminary data, in both groups, deterioration was observed during the period of orthodontic treatment. In both groups, 34 patients (57.6%), 176 teeth (21.6%), had demineralization phenomena in the form of white spots. A comparative analysis revealed a certain difference in the degree of damage in the control and prophylactic groups in favor of the prophylactic group, where the statistical significance was *p* ≤ 0.01.

In the group with patients aged 12–14 years (mean age 12.88 ± 0.83), white spots formed in 20 patients (62.5%) on 112 teeth (23.3%). On average, 3.5 ± 3.2 teeth were affected in one patient in this group. In the control group, white spots were detected on 68 teeth (60.7%). There were more white spots on the anterior teeth—49 (72.1%) and fewer on the posterior teeth of the lateral group—19 (27.9%) (t = 5.2; *p* ≤ 0.0001). In the prophylactic group, 44 teeth (37.7%) were affected, of which 31 were incisors (70.5%) and 13 were molars (29.5%) (t = 3.48; *p* ≤ 0.002). The difference between the identified foci of demineralization in the control and prophylactic groups of the patients aged 12–14 years was statistically significant (t = 4.98; *p* ≤ 0.001). In the group of patients aged 15–17 years (mean age 15.96 ± 0.8), white spots were detected in 14 people (51.9%) on 64 teeth (19.05%). On average, 2.37 ± 2.8 teeth were affected in one patient in this group. In the control group, white spots were detected on 41 teeth (64.06%). Likewise, more on the incisors—29 (70.7%), and fewer on the posterior teeth of the lateral group—12 (29.3%) (t = 3.25; *p* = 0.003). In the prophylactic group, 23 teeth (35.9%) were affected, of which 13 were incisors (56.5%) and 10 were molars (43.5%) (t = 1.1; *p* = 0.265). The difference between the detected foci of demineralization in the control and prophylactic groups of the patients aged 15–17 years was statistically significant (t = 2.449; *p* = 0.021). In the control groups (C, E), no white spots were found upon initial examination. Upon final examination, the number of spots was 109 (61.9%). In the prophylactic groups (D and F) over the same time period, the number of white spots was 67 (38.1%) (t = 5.3; *p* ≤ 0.001). As a result of observation, the number of teeth with white spots in children aged 12–14 years was more (63.6%) than the index of the age group of 15–17 years (36.4%) in all the mentioned groups (C, D, E, and F).

When analyzing these indicators of laser fluorometry, the following results were obtained. Before the start of orthodontic treatment in the group of 12–14-year-old patients, the mean values of KaVo DIAGNOdent were 9.69 ± 2.48 (*p* ≤ 0.0001). After 6 months of the orthodontic treatment and preventive measures with Tiefenfluorid, in the prophylactic group, the average fluorometry values increased to 13.66 ± 3.67 (*p* ≤ 0.0001). In the control group, the average fluorometry values increased to 15.75 ± 5.4 *p* (*p* ≤ 0.0001). In patients with white spots in group D, the indices were 16 ± 2.34, and in group C, they were 19.45 ± 2.98. The difference in indicators was statistically significant (t = 12.96; *p* ≤ 0.0001).

Before the startup of orthodontic treatment in the group of 15–17-year-old patients, the mean values of KaVo DIAGNOdent were 13.52 ± 2.48 (*p* ≤ 0.0001). After 6 months of the orthodontic treatment and prophylactic measures with Tiefenfluorid, the mean fluorometry values increased to 13.33 ± 4.3 in the prophylactic group and to 14.67 ± 5.05 in the control group (*p* ≤ 0.0001). In the patients with white spots in group F, the indices were 16.79 ± 3.04, and in group E, the indices were 18.93 ± 2.79. The difference in indicators was statistically significant (t = 6.51; *p* ≤ 0.0001).

## 4. Discussion

The analysis of the obtained results of the experiment confirmed parallel clinical observations of the same patients using the split-mouth technique. This created equal conditions for both groups, and ensured statistical significance. The indicators of the prevalence of white spots in patients receiving orthodontic treatment were different. In patients in the 12–14-year-old age group, the foci of demineralization were found more often (58.8%). This can also be explained by the incomplete mineralization of tooth enamel at this age [9,10,31].

Non-removable orthodontic appliances create additional points for the accumulation of food debris, increasing the surface area to which microbial biofilm can attach. Due to their irregular shape, it is almost impossible to completely remove microbial biofilm. As a result, one can infer that orthodontic treatment is a risk factor for developing white spots. As a rule, white lesions due to orthodontic treatment occur on the buccal surface of the teeth, and are difficult to clean. This is the area between the brackets and the gum, especially in the upper lateral incisors and canines, as well as in the lower canines and first premolars. This is supported by the data of some investigators who noted the development of white spots in 15 to 60% of cases [8,32]. These lesions can develop very rapidly over 4 weeks, affecting deeper areas, making them impossible to re-calcify. Therefore, it is very important to identify these lesions as quickly as possible [14]. Unfortunately, once these lesions become clinically visible, the changes become irreversible.

Clinical studies have confirmed that the regular use of a prophylactic agent, which does not depend on patient compliance during the orthodontic treatment, reduced the risk of a cariogenic situation by 38.07%. Protection of the tooth surface during orthodontic treatment is significant, considering the high incidence of enamel mineral loss that occurs during non-removable appliance treatment at aesthetically important sites, such as the vestibular surfaces of maxillary incisors [4,29,33]. The study showed that in 69.3% of cases, the lesions were localized on the anterior teeth. The prevalence of foci of demineralization on molars was 30.7%, which can be explained by the peculiarity of fixing braces on this group of teeth (the use of bondage rings, which occupy a significant surface area and so accumulate even more microbial biofilm on their surface). In addition, the patients face difficulties in brushing this group of teeth due to their location [3].

An in vitro study showed that the use of a prophylactic agent around the base of the bracket leads to a decrease in the depth of enamel demineralization by 1.8 times, confirming the hypothesis of the study. The decrease in depth is predetermined by two-factor action: Tiefenfluorid in combination with highly dispersed calcium hydroxide realizes deep fluoridation of tooth enamel by the formation of submicroscopic CaF_2_ crystals. Packed in a gel and thus protected from mechanical influences, these crystals constantly emit fluorine, which, in turn, together with the mineral salts of saliva provide long-term remineralization, enhancing the fluorine by almost 100 times [19,20]. The protective properties of fluorine, which have been absorbed onto the surface of the enamel, disappear over time. This is why coating with Tiefenfluorid was carried out every 6 months.

During the study of histological sections, enamel decalcification was detected. There is a difference between the thicknesses of the layers, which indicates more pronounced demineralization processes in one group rather than in the other. Since the selection of teeth for groups was random, this difference cannot be a hidden or obvious reason for demineralization (*p* ≤ 0.001). A limitation of the study is that, in the experimental groups, the extracted teeth under examination were not located in their natural medium; there was no natural nutrition of the teeth. This limitation was compensated by the circumstances in which the experimental group was compared with the control group, where the teeth were processed by the same methods, in the most approximate conditions. In vitro studies allow one to study sections of teeth, which enable factual description of the formation of demineralization, as well as measuring the depth of the affected area.

## 5. Conclusions

The obtained data and evaluation of the local fluoridation method using the enamel-sealing liquid Tiefenfluorid for deep fluoridation determined that Tiefenfluorid has powerful bactericidal activity and helps to suppress the formation of a microbial biofilm on the enamel surface. This is due to the inclusion of copper ions in the composition, which serve as an incentive to develop alternative methods for increasing the resistance of enamel. Thus, further studies of this new and promising solution are needed.

## Figures and Tables

**Figure 1 ijerph-20-00540-f001:**
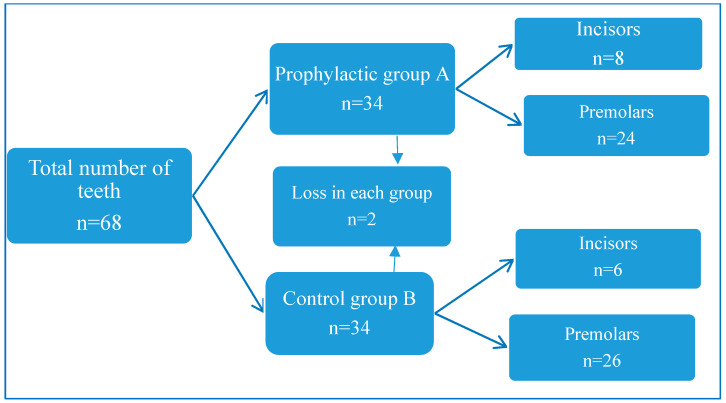
Representation of the distribution of groups and subgroups of teeth in this study.

**Figure 2 ijerph-20-00540-f002:**
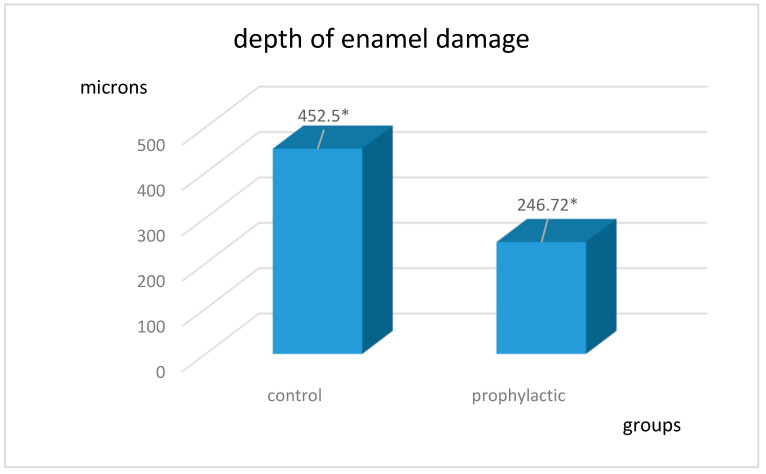
Depth of enamel damage in the control and prophylactic groups in microns. * The results obtained are statistically significant: *p* < 0.0001.

**Figure 3 ijerph-20-00540-f003:**
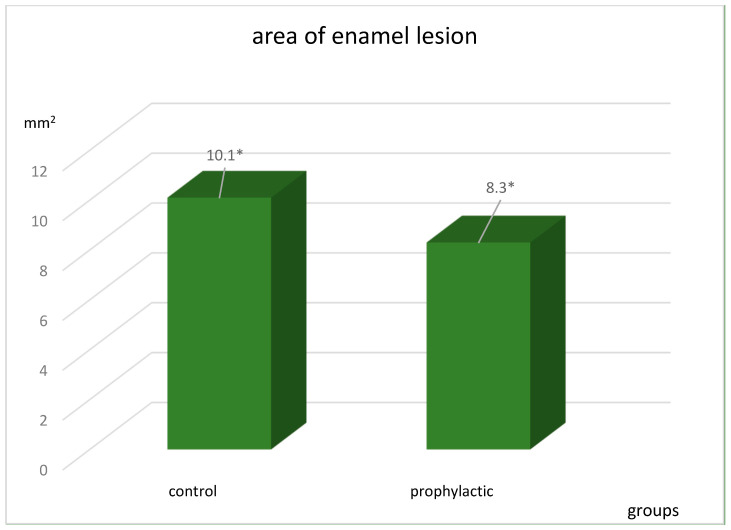
Area of enamel lesion in the staining stage in the control and prophylactic groups in mm^2^. * The results obtained are statistically significant: *p* < 0.0001.

**Figure 4 ijerph-20-00540-f004:**
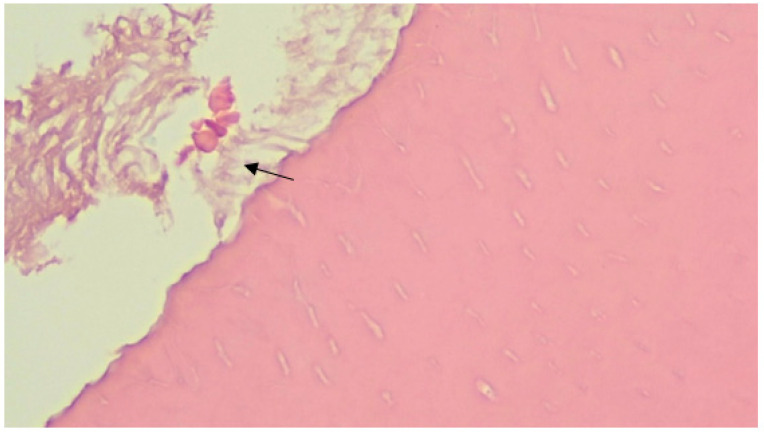
The focus of deep demineralization of tooth enamel in the control group (microscopic examination of the enamel in the area of the carious spot revealed changes in the crystals: a violation of the orientation of the crystals in the structure of hydroxyapatites, a change in the shape of the crystals, their size, and the appearance of crystals atypical for normal enamel. Staining with Ehrlich’s hematoxylin, in magnification ×200).

**Figure 5 ijerph-20-00540-f005:**
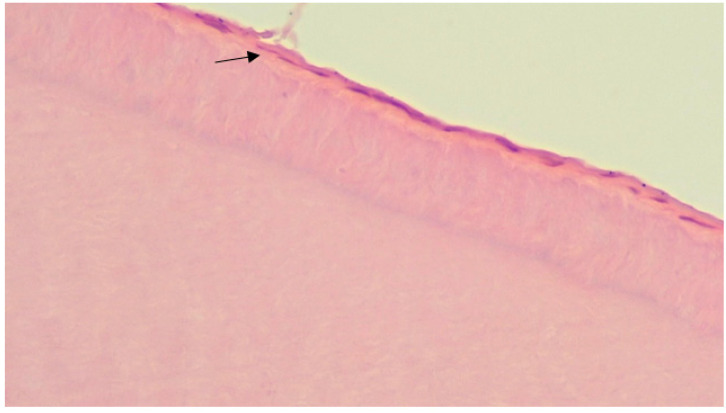
Superficial enamel lesion in the control group (during microscopic examination of enamel, changes develop only in the surface layers of enamel in the form of displacement of enamel prisms relative to each other. Staining with Ehrlich’s hematoxylin, in magnification ×200).

## Data Availability

Not available.

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
