# Peer review of "Morphological Characteristics and Prevention of Tooth Enamel Demineralization during Orthodontic Treatment with Non-Removable Appliances"

_ijerph, 2022, doi:10.3390/ijerph20010540_

Round 1
Reviewer 1 Report
Dear authors,
The English you use has to pass through significant changes. Maybe you were writing the article in a hurry. I advise you to consult an English native speaker.
In Methods, please specify which paste you use for brushing teeth for 5 seconds.
Please explain why you choose Tiefenfluoride as material.
What is the reason to brush the teeth twice a day as long as you have performed an ex vivo study?
Please describe the caries-stimulating solution.
All this section is hard to read and understand.
Discussions
At 12-14 years the teeth have completed the mineralization.
White spots are not impossible to calcify. ICON has been proven to get very good results in this matter.
It is a big difference between the incisors’ and premolars’ demineralization at the same age.
Author Response
Please, see the attachment

Reviewer 2 Report
Dear Authors the paper is really interesting, well conducted and fits the objectives of the journal; but it is necessary to review some points in order to improve the quality of the paper:
-First, i ask you to check the plagiarism of your article using specific sites to get a similitary report
- The introduction section is very short and is needed to add other references to increase the quality of the manuscript
I suggest you some articles that will help you improve your article.
A new combined protocol to treat the dentin hypersensitivity associated with non-carious cervical lesions: A randomized controlled trial DOI 10.3390/app11010187
Dento-Skeletal Class III Treatment with Mixed Anchored Palatal Expander: A Systematic Review DOI: 10.3390/app12094646
Patient-reported outcomes while managing obstructive sleep apnea with oral appliances: a scoping review. Journal of Evidence-Based Dental Practice. 2022 Oct;101786. Doi: https://doi.org/10.1016/j.jebdp.2022.101786
-You need to review the grammar and English of your article, with the help of a native English speaker (you can specify who helped you in reviewing English in the acknowledgements) or simply by using a site that can support you in English
-I suggest you to add an image in order to improve the iconography of the article.
-Please expand conclusion section with main results and future perspectives of this study
Regards
Author Response
Please, see the attachment.
